# Update on the Role of Allergy in Pediatric Functional Abdominal Pain Disorders: A Clinical Perspective

**DOI:** 10.3390/nu13062056

**Published:** 2021-06-16

**Authors:** Craig Friesen, Jennifer Colombo, Jennifer Schurman

**Affiliations:** Division of Gastroenterology, Hepatology and Nutrition, Children’s Mercy Kansas City, School of Medicine, University of Missouri-Kansas City, Kansas City, MO 64108, USA; jmcolombo@cmh.edu (J.C.); jschurman@cmh.edu (J.S.)

**Keywords:** functional abdominal pain disorders, functional dyspepsia, irritable bowel syndrome, food allergy

## Abstract

Both functional abdominal pain disorders (FAPDs) and food allergies are relatively common in children and adolescents, and most studies report an association between FAPDs and allergic conditions. FAPDs share pathophysiologic processes with allergies, including both immune and psychological processes interacting with the microbiome. No conclusive data are implicating IgE-mediated reactions to foods in FAPDs; however, there may be patients who have IgE reactions localized to the gastrointestinal mucosa without systemic symptoms that are not identified by common tests. In FAPDs, the data appears stronger for aeroallergens than for foods. It also remains possible that food antigens initiate an IgG reaction that promotes mast cell activation. If a food allergen is identified, the management involves eliminating the specific food from the diet. In the absence of systemic allergic symptoms or oral allergy syndrome, it appears unlikely that allergic triggers for FAPDs can be reliably identified by standard testing. Medications used to blunt allergic reactions or symptomatically treat allergic reactions may be useful in FAPDs. The purpose of the current manuscript is to review the current literature regarding the role of allergy in FAPDs from a clinical perspective, including how allergy may fit in the current model of FAPDs.

## 1. Introduction

Chronic or recurrent abdominal pain is common in children and adolescents, with a worldwide prevalence estimated at 13.5% [1]. Most youth with chronic abdominal pain will fulfill the functional abdominal pain disorder (FAPD) criteria as defined by the Rome criteria [2]. The recognized FAPDs include irritable bowel syndrome (IBS), functional dyspepsia (FD), abdominal migraine, and functional abdominal pain, with IBS and FD being the two most common [2,3,4].

There has been increasing interest in the role of diet in FAPDs. Perceived food intolerances are common in pediatric FAPD patients, with over 90% identifying at least one food they associate with worsening symptoms [5]. These patients frequently avoid specific foods and self-implement dietary strategies [5]. There are a variety of mechanisms by which specific foods can increase symptoms, including food allergy (immunologic reactions), food intolerances (non-immunologic reactions, e.g., malabsorption), and reactions created by hypervigilance and anticipation of symptoms in patients with perceived intolerances that may increase anxiety with consumption of the suspected food [6,7]. Perceived intolerances may also be influenced by underlying psychological factors [8].

The separation between food allergy and intolerance has become increasingly blurred as some food intolerances can start a chain of events resulting in mucosal immune activation. For example, lactose malabsorption is a well-recognized food intolerance. In most studies, lactose restriction does not result in clinical improvement even in patients with demonstrated malabsorption [9,10,11,12]. It is now recognized that this malabsorption is associated with increased mucosal mast cells and increased colonic eosinophils and lymphocytes, which may persist after lactose elimination [13,14,15,16]. Non-absorbed sugars and fructooligosaccharides alter the intestinal microbiome and production of short-chain fatty acids, both of which affect the development of food allergies [6,13,14,17,18,19,20]. The altered microbiome can interrupt the intestinal epithelial barrier (another factor highly implicated in FAPDs) with a subsequent increase in the immune system’s exposure to luminal food and microbial antigens [21]. Lastly, it is recognized that dietary compounds (or metabolic byproducts) can modulate mast cell function [22]. For example, the benefits of fiber supplements are in part due to slow fermentation, producing short-chain fatty acids which preserve the intestinal barrier and decrease inflammation, including inhibition of MC activation [23,24,25].

Food allergy refers to developing symptoms resulting from an immune reaction (generally involving mast cells and eosinophils) to an ingested antigen. Food allergies are divided into two categories: IgE- mediated and non-IgE-mediated. IgE-mediated reactions are associated with more rapid onset of symptoms, while non-IgE-mediated reactions typically result in delayed onset of symptoms [26,27]. Food allergy in children has an estimated worldwide prevalence of 6–8%, with estimates of 10% in high-income countries [27]. Approximately 50% of food allergy reactions will produce systemic symptoms (e.g., wheezing, hives, anaphylaxis), and 50% will produce only or primarily gastrointestinal symptoms [26]. Multiple physiologic factors prevent immune reactions to foods, termed tolerance, including microbiome features and the intestinal barrier [28]. Importantly, “outgrowing” a food allergy is associated with the development of food-specific IgG rather than IgE [29].

Both FAPDs and food allergies are relatively common in children and adolescents and may be linked in at least a subset of patients. The purpose of the current manuscript is to review the current literature regarding the role of allergy in FAPDs from a clinical perspective, including how allergy may fit in the current model of FAPDs. Although the focus is on pediatric FAPDs, we will also incorporate the more abundant adult literature relevant to adolescents.

## 2. Inflammation and the Biopsychosocial Model

The complex nature of chronic abdominal pain has long been viewed within the context of the biopsychosocial model, which recognizes that various interacting factors contribute to the initiation and maintenance of pain. These contributors include biologic factors (e.g., genetics, visceral hypersensitivity, inflammation, dysbiosis), psychologic factors (e.g., anxiety, depression), and social factors (e.g., poor relationships with parents, teachers, or peers). There appear to be four main host systems involved in symptom generation which interact readily with each other, including psychologic, neurologic, immunologic, and endocrinologic systems, all of which interact with the gastrointestinal microbiome. A central mechanism appears to be visceral hypersensitivity, an exaggerated response to stimuli such as gastrointestinal distension. Hypersensitivity to distension has been demonstrated in youth with chronic abdominal pain [30,31,32].

Given that allergy involves an immunologic reaction, the role of inflammation, particularly mast cell-related, within the biopsychosocial model appears to be most relevant to the current discussion (See Figure 1). Mast cells are generally positioned at interfaces between the host and environment, providing a connecting link between the neurologic and immunologic systems and, in part, a link between the enteric and central nervous systems [33]. As will be discussed, mast cells are also an important link to the psychologic system. 

There is considerable evidence implicating inflammation in FAPDs, particularly mast cells (and to a lesser degree, eosinophils) in IBS and both mast cells and eosinophils in FD [34,35,36,37,38]. With activation, both mast cells and eosinophils release mediators with biologic effects relevant to FAPDs. These mediators can stimulate afferent nerves sending a pain signal, sensitize afferent nerves inducing visceral hyperalgesia, and alter electromechanical function [36]. Mast cells are highly implicated in IBS, with increased density reported in the colon and ileum in over 80% of published studies investigating this relationship [34,39]. In addition, IBS is associated with an increased density of degranulating mast cells and mast cells in proximity to nerves which correlate with abdominal pain frequency and severity [40]. Both mast cell and eosinophil densities have been shown to be increased in pediatric IBS [41]. Increased densities of both mast cells and eosinophils have been demonstrated in FD, as has increased activation of mast cells and/or eosinophils in both adults and youth with FD [35,42,43,44]. Mast cell degranulation in the proximal stomach may be associated with visceral hyperalgesia in adults with FD [45]. To what degree mucosal inflammation may result from allergic reactions is unclear, but a history of allergy has been associated with increased duodenal eosinophils in adults with FD [46]. In children with cow’s milk allergy, mucosal application of milk results in increased mast cell and eosinophil density and activation, as well as an increase in mast cells in proximity to nerves, findings similar to those reported in FAPDs in general [47].

Psychologic function is an important component of the biopsychosocial model and may directly relevant to a discussion of allergy. FAPDs are associated with psychologic dysfunction, including anxiety, depression, and maladaptive coping [48]. Psychologic disturbances are associated with greater abdominal pain severity and predict worse outcomes and persistence into adulthood [48,49,50]. In addition, asthma, allergic rhinitis, atopic dermatitis, and food allergy are associated with increased stress, changes in mood, and emotional dysfunction [51,52]. Psychologic functioning interacts with biologic functioning (in a bi-directional fashion) particularly with the immunologic system; anxiety and depression have been associated with increased mast cell and/or eosinophil density in youth and adults with FAPDs [41,53,54,55]. Psychologic functioning also interacts with the endocrinologic system as anxiety can trigger a stress response initiated by corticotropin-releasing hormone (CRH) release. Through activation of CRH receptors, stress results in mast cell degranulation, disrupting the epithelial barrier, increasing antigenic exposure [56]. As stress can exacerbate symptoms, its presence can also create difficulty determining a patient’s response to food restrictions. Santos and colleagues studied a group of adults with documented food allergies [57]. Under conditions of cold stress, these patients exhibited increased luminal release of tryptase and histamine in the jejunum at a magnitude comparable to that induced by food allergen exposure [57]. Thus, a patient with food allergies may have symptoms triggered by other factors even after eliminating the food allergen. While allergies may have a role in FAPDs in at least a subset of patients, it is important to recognize the complex nature of chronic abdominal pain and the other factors that may be active in symptom generation.

## 3. Allergy and Functional Abdominal Pain Disorders

Most, but not all, studies have shown an association between FAPDs and allergic conditions [58]. FD and functional abdominal pain have been associated with asthma in adolescents [58]. Likewise, asthma and food allergy, allergic rhinitis, and eczema have been associated with FAPDs in adults [59,60,61,62,63,64]. In an extensive primary care study, both FD and IBS were associated with allergic conditions, and the relationship was partially explained by a common association with anxiety and depression [63]. Allergic conditions early in life also appear to predispose to later development of FAPDs. Pre-schoolers with allergic disease have an increased risk of IBS when they reach school-age, with earlier development of IBS in those with food allergies [65]. The highest risk was associated with allergic rhinitis [65]. In another study, the risk of childhood IBS was significantly increased in those with a history of atopic dermatitis [66]. Allergic proctocolitis early in life is also a risk factor for subsequent FAPD development [67]. The association of allergies early in life and subsequent FAPD development is not well understood. Still, there is some evidence that allergy may alter the microbiome, which could predispose to FAPDs, or allergy, and FAPD could be epiphenomena related to dysbiosis. Children with food sensitization have lower microbiota diversity overall with lower Bacteroides and higher Firmicutes colonization [18]. The microbial signature can distinguish between those with IgE-mediated and those with non-IgE-mediated food allergies in infants [19].

Interestingly, a placebo-controlled trial of a probiotic (Bifidobacteria) in children with allergic rhinitis in association with intermittent asthma demonstrated significant improvement in allergic rhinitis symptoms [68]. These findings suggest shared pathophysiology related to the gastrointestinal microbiome. Lastly, bacteria-derived (and host-derived) proteases have been implicated in disruption of the intestinal barrier, increasing antigen exposure, and may also directly stimulate mast cells and sensory neurons [69].

### 3.1. IgE-Mediated Allergies

No conclusive data is implicating IgE-mediated reactions to foods in FAPDs [70,71]. The gold standard for diagnosis of food allergy is a double-blind, placebo-controlled food challenge that can be time-consuming and are most likely to be helpful in FAPD patients who also experience systemic reactions with food ingestion [72]. The only proven clinically utilized diagnostic techniques for IgE-mediated reactions are skin prick tests (SPT), which have high sensitivity and low specificity, and measurement of serum food-specific IgE, which also have low specificity [72]. Both tests are indicative of sensitization but by themselves are not diagnostic of clinical allergy. The low specificity of food-specific IgE can be particularly problematic when ordering large panels. In a study of 220 adults with IBS and/or FD, food-specific IgE tests were positive in 38% [73]. On an elimination diet, a positive response was seen in 8 of 19 patients, all of whom relapsed on reintroduction, yielding an overall prevalence of 4% for IgE-medicated food allergy [73]. Not only is this frequency similar to that seen in the general population, but the study highlights the limited ability of a positive test to predict clinical symptoms. Another study of adults with FD and IBS found no differences in food-specific IgE compared to controls [74]. We previously found no increase in immunoreactivity (including IgE, SPT, IgG, IgG4, and atopy patches) to common food allergens in children with FD and duodenal eosinophilia [75].

There may be patients who have IgE reactions localized to the gastrointestinal mucosa without systemic symptoms who are not identified by SPT or serum food-specific IgE. Methods are available to evaluate localized reactions in the gastrointestinal tract, including the colonoscopic allergen provocation test (COLAP; mucosal testing akin to the SPT) and visualization utilizing confocal laser endomicroscopy (CLE). Utilizing COLAP, positive reactions to food antigens are recognized by the wheal and flare reactions occurring within 20 min of antigen application. Reactions are associated with mast cell degranulation eosinophil activation histologically [76]. In a study of 70 adults with gastrointestinal symptoms suspected to be related to food allergy, COLAP was positive in 97/210 (46%) of challenges in patients and not in any challenges in controls [76]. Reactions correlated with patient histories of food reactions but not SPT results or food-specific IgE [75]. In sum, these findings may indicate a higher rate of IgE-mediated food allergy in FAPD patients and cast some doubt on the sensitivity of standard allergy tests and their ability to rule out allergies to specific foods in the absence of systemic symptoms. 

Recent studies in a mouse model demonstrate potentially important interactions with bacterial infection or colonization in predisposing to these localized intestinal allergic reactions [77]. In this model, the bacterial infection causes a loss of oral tolerance resulting in mucosal food-specific IgE and increased visceral pain via IgE- and mast cell-mediated mechanisms [77]. Studies in this model also demonstrate a possible role for superantigens, which are microbes known to cause non-specific activation of T lymphocytes and which have been implicated in non-gastrointestinal atopic conditions. The primary superantigens, *Staphylococcus aureus* and *Streptococcus pyogenes*, are more commonly present in the microbiome of IBS patients [77]. Skin colonization with *Staphylococcus aureus* in patients with atopic dermatitis has been associated with an increased risk of food allergy [78]. Likewise, a loss of balance and diversity in the intestinal microbiome increases food allergy risk [21].

In FAPDs, the data appears stronger for aeroallergens than for foods [79]. As discussed previously, FAPD is associated with allergic rhinitis, and allergic rhinitis early in life increases FAPD risk. In addition, adults with IBS have an increased risk of seasonal allergies (and consequently pollen-food syndrome), and seasonal allergic rhinitis is associated with greater IBS severity [61]. Aeroallergens enter the nasal and oral cavities with breathing and may be swallowed, or they may be ingested following food contamination [58]. In children with FAPDs, local pollen counts are associated with the onset of pain and are as strong a predictor as are affect or sleep disturbances [80]. Birch pollen, in particular, has been well studied. During birch pollen season, adults with birch pollen allergy demonstrate an increase in duodenal eosinophils and IgE-carrying mast cells along with oral allergy syndrome [81]. Oral allergy syndrome is a hypersensitivity to raw plant proteins, often proteins that cross-react with pollen proteins, resulting in oropharyngeal symptoms (e.g., itching, tingling, swelling) [26]. In a separate study, birch exposure was associated with increased intestinal eosinophil and mast cell densities [82]. Patients with gastrointestinal symptoms had increased IgE to birch (rBet v 1), hazelnuts, and apple [82]. In a study of patients with birch pollinosis, COLAP with rBet v 1, a positive reaction was seen in 81% where there was a history of gastrointestinal symptoms. There were 22% of those with pollinosis and no gastrointestinal symptoms and none in the healthy controls [83]. Aeroallergens might not only be a trigger for symptoms in FAPD patients but an indicator of potential food triggers.

### 3.2. Non-IgE-Mediated Allergies

Food antigens may precipitate gastrointestinal symptoms through cell-mediated processes (type IV hypersensitivity reactions [84]. The classic examples include food protein-induced allergic proctocolitis (FPIAP), a benign condition generally presenting in early infancy, and food protein-induced enterocolitis syndrome (FPIES), generally presenting with severe symptoms within the first 6 months of life [85]. While not completely characterized, FPIES results from food-induced immune activation, including activation of the innate immune system limited to the gastrointestinal tract [86,87]. FPIES is associated with disruption or a lack of development of tolerance [84]. FPIES most often presents with severe bouts of abdominal pain, nausea, vomiting, and diarrhea, symptoms also seen in FAPDs. While FPIES most often resolves by one year of age, it can persist into or develop during adolescence or adulthood [84,87,88,89]. Reactions in older children, adolescents, and adults are most frequently described in relation to seafood ingestion, but it is possible that milder cell-mediated reactions to other foods could contribute to FAPDs [87,88,89]. Increased density and activation of T lymphocytes and indirect evidence for TH17 activation have been demonstrated in adults with FAPDs [38]. While TH17 cells may have a pro-inflammatory role, they may also serve a protective role, depending on the inflammatory milieu, microbiome composition, and epigenetic modifications [90,91,92,93,94]. Under specific conditions, TH17 cells can induce eosinophil infiltration and activation and, potentially, mast cell accumulation [95,96,97,98]. Increased mucosal TH17 density has been demonstrated in pediatric IBS and pediatric FD associated with chronic gastritis [41,99]. In FD associated with chronic gastritis (but not in the absence of chronic gastritis), gastric and duodenal TH17 density was greater than controls and comparable to that seen in Helicobacter-pylori-associated gastritis and Crohn’s-associated gastritis [99]. Lastly, in another study utilizing confocal laser endomicroscopy (CLE), 155 adults were challenged with four common food antigens, all of which were negative on SPT and without elevations of specific IgE. However, a localized mucosal IgE reaction was not assessed [100]. Of the 108 completers in the study, 70% had a positive test, and patients with positive tests were 4X more likely to have another atopic disorder than controls [100]. A positive response was associated with increased permeability, increasing mucosal lymphocyte density, and eosinophil (but not mast cell) activation [100]. Thus, there appears to be a role for lymphocytes in immunologic food reactions in the absence of IgE secretion. 

There have been multiple studies, primarily in adults, evaluating food-specific IgG in patients with FAPDs. At present, IgG testing is discouraged by most major allergy organizations as it lacks proven clinical utility [101]. It is believed to be indicative of exposure, and as previously discussed, tolerance to a specific food is associated with the development of food-specific IgG [29,102]. However, the utility of food-specific IgG testing specifically in FAPDs awaits further evaluation, and, although not proven, there is data suggesting a potential for clinical utility. Multiple studies have reported increased food-specific IgG or IgG4 in adults with FAPD, particularly IBS [43,74,101,103,104]. Multiple studies have also reported clinical improvement on elimination diets guided by IgG or IgG4 testing [104,105,106,107,108,109]. However, only one of these studies restricted foods in a blinded fashion, and there were some other significant differences between the true and sham restricted diets in this study [107]. However, a greater benefit was seen with better compliance [107]. Another study found improvement in both compliant and non-compliant adults [106].

Additionally, while IgG titers are not related to symptom severity, they correlate with mast cell density and degranulation [74,108]. It remains possible that increase antigen exposure through an impaired epithelial barrier instigates a food-specific IgG reaction that promotes mast cell activation. However, further studies are needed before IgG testing should be considered. 

Atopy patch testing is a diagnostic procedure for identifying delayed hypersensitivity reactions that have been primarily useful in identifying allergens in contact dermatitis [72]. In very limited, uncontrolled studies of adults with IBS, patch testing has been reported to identify foods that, when restricted, resulted in clinical improvement [110,111]. There is currently insufficient data to support patch testing in FAPDs.

## 4. Management

If a food allergen is identified, allergy management is relatively straightforward and involves eliminating the specific food from the diet. However, it appears quite unlikely that many patients with FAPDs will benefit from a standard approach that includes routine allergy testing to identify the cause of symptoms which, when eliminated from the diet, will result in the resolution of the FAPD. More likely, indiscriminate testing, particularly utilizing large panels identifying food-specific IgE (or IgG) elevations, will result in unnecessary diet restrictions without long-term benefit and could lead to nutritional deficiencies. In patients with only gastrointestinal symptoms, allergy testing is not likely to be helpful, either in identifying the culprit or in providing a list of safe foods. Patients who develop systemic allergic symptoms or those with oral allergy syndrome should be referred to an allergist for directed testing and management of identified allergens. This should not be a transfer of care but the formation of a collaboration to manage the patient. Another approach is an empiric restriction of the most common allergens used in the treatment of eosinophilic esophagitis. Unfortunately, there is very little data assessing this approach in patients with eosinophilic gastroenteritis, let alone FAPDs, and no randomized trials [112,113]. Institution of highly restrictive diets is not without risks, including decreased quality of life and potential nutritional deficiencies, and should only be instituted long-term in collaboration with a dietician [114]. 

While the patient would be expected to benefit from removing food allergens or managing aeroallergens if identified, it remains unlikely that gastrointestinal symptoms will completely resolve without addressing other biopsychosocial aspects of these disorders as allergens are only one trigger for activating gastrointestinal mast cells. See Figure 2 for an overview of therapeutic targets within FAPD pathophysiology. Anxiety/stress and depression may be important therapeutic targets as both FAPDs and allergies are associated with psychologic dysfunction, triggering symptoms. Treatment of stress has been shown to benefit allergic conditions and in children with FD in association with duodenal eosinophilia [52,115].

For patients with primarily gastrointestinal symptoms who do not have systemic symptoms typical of allergic reactions or oral allergy syndrome, it is unlikely that specific triggering allergens will be identified even if they exist as current tests likely lack adequate sensitivity or specificity. The challenge is that end-organ inflammatory processes do not necessarily identify patients who specifically have allergic triggers, as mast cells and eosinophils appear to be a common component of FAPDs. Another approach would be to direct treatment at the mast cells and/or eosinophils without regard to whether their activation is due to an allergen. As with non-gastrointestinal allergic diseases, treatment can be directed at mast cell activators (e.g., anti-IgE, anti-IL-5), mast cell mediator release (e.g., mast cell stabilizers, anti-siglec-8), or inhibition of mast cell mediators (e.g., histamine and cys-leukotrienes) at their effector sites [36]. Most have been studied in the context of FAPDs or mucosal eosinophilia with some data in patients with demonstrated food allergy.

There are no FDA-approved drugs for the treatment of gastrointestinal eosinophils [116]. Multiple biologics have been developed or are in development, which targets upstream mediators associated with mast cell and eosinophil infiltration and/or activation [See Pesek [116] and Wechsler [117] for a full review]. The most studied, in general, are omalizumab, a monoclonal antibody directed at IgE, and mepolizumab and reslizumab, monoclonal antibodies directed at interleukin 5. Omalizumab has demonstrated efficacy in providing a degree of protection from peanut allergy [118]. It has also been studied in 9 patients with FAPD symptoms and mucosal eosinophilia [119]. It was associated with symptom improvement, but decreases in antral and duodenal eosinophil densities did not reach statistical significance; the study was likely underpowered for assessment of changes in eosinophil density [119]. Anti-interleukin 5 antibodies have not been studied in eosinophilic gastroenteritis. Currently, stress appears to be the most viable upstream treatment target.

The largest body of data, albeit somewhat meager, exists for mast cell stabilizing medications, including oral cromolyn and ketotifen. Two placebo-controlled trials have assessed the response to ketotifen, a mast cell stabilizer and H1 antagonist, in adults with IBS, with both demonstrating improvement in abdominal pain and other IBS symptoms [120,121]. Visceral hypersensitivity improved in both while decreases in mast cell density and activation were seen in only one study [120,121]. Whether effects resulted from mast cell stabilization or an antihistamine effect, or both is not clear. Oral cromolyn has been studied in patients with IBS, frequently associated with positive allergy testing, and, in comparison to restricted diets [122,123,124,125,126]. There have been two pediatric studies [122,123]. In a study of children with abdominal pain and diarrhea, cromolyn was shown to be as effective as diet restriction guided by SPT [122]. A study of 10 children with egg allergy found no benefit from cromolyn in preventing egg reactions [123]. A study of 20 adults with IBS and documented food intolerances found that oral cromolyn allowed patients to tolerate their offending foods [124]. Another study in adults found oral cromolyn to be comparable to an elimination diet, with both treatments performing better in patients with positive food SPTs [125]. Lastly, another study in adults found oral cromolyn to improve mast cell activation, abdominal pain, and stool consistency [126]. A newer biologic targeting siglec-8 (which is present only on mast cells and eosinophils) has been shown to eliminate eosinophils and inactivate mast cells [127]. It has shown positive results in phase II trials, decreasing antral and duodenal eosinophils and improving symptoms [127]. Prevention of mast cell activation may be a viable strategy in FAPDs, including possibly, patients with food allergies who do not exhibit systemic allergic symptoms.

Medications used in other allergic conditions to inhibit the actions of mediators released from mast cells and eosinophils may also be beneficial in FAPDs, although the current evidence is limited. (See Table 1) Most studies have been undertaken in patients with mucosal eosinophilia or increased mucosal mast cells. Histamine has received significant attention as it has been shown to sensitize TRPV1 receptors, promoting visceral hypersensitivity [128]. H1 antagonists alone or in combination with H2 antagonists have been reported to be effective in children and adults with FD in uncontrolled studies [129,130,131]. In adults, the response was predicted by elevated duodenal eosinophils [131]. Ebastine, an H1 antagonist, has decreased visceral sensitivity in adults with IBS [128]. Another potential downstream target is cysteinyl- leukotriene (cysLT) receptors. Montelukast, a cysLT receptor antagonist, effectively treats pain in children with FD and duodenal eosinophilia. However, the effect is independent of changes in mast cell or eosinophil density or activation [132,133]. While not an anti-allergy drug, per se, proton pump inhibitors have proven efficacy in eosinophilic esophagitis and may be effective for treating duodenal eosinophilia [134,135,136]. In a prospective study of adults with FD and duodenal eosinophilia, treatment with pantoprazole was associated with improved symptoms and reduced mucosal eosinophil density [136].

## 5. Conclusions

FAPDs share pathophysiologic processes with allergies, including both immune and psychological processes interacting with the microbiome. Although there is significant overlap in the medications that can be used for allergic disorders and FAPDs (particularly IBS and FD in association with mucosal eosinophilia), it is unclear to what degree allergens play a role FAPDs. In the absence of systemic allergic symptoms or oral allergy syndrome, it appears unlikely that allergic triggers for FAPDs can be reliably identified by standard testing. There is a need for high-quality studies assessing dietary strategies and anti-allergy medication to better understand the efficacy and the value of allergy tests to predict response.

## Figures and Tables

**Figure 1 nutrients-13-02056-f001:**
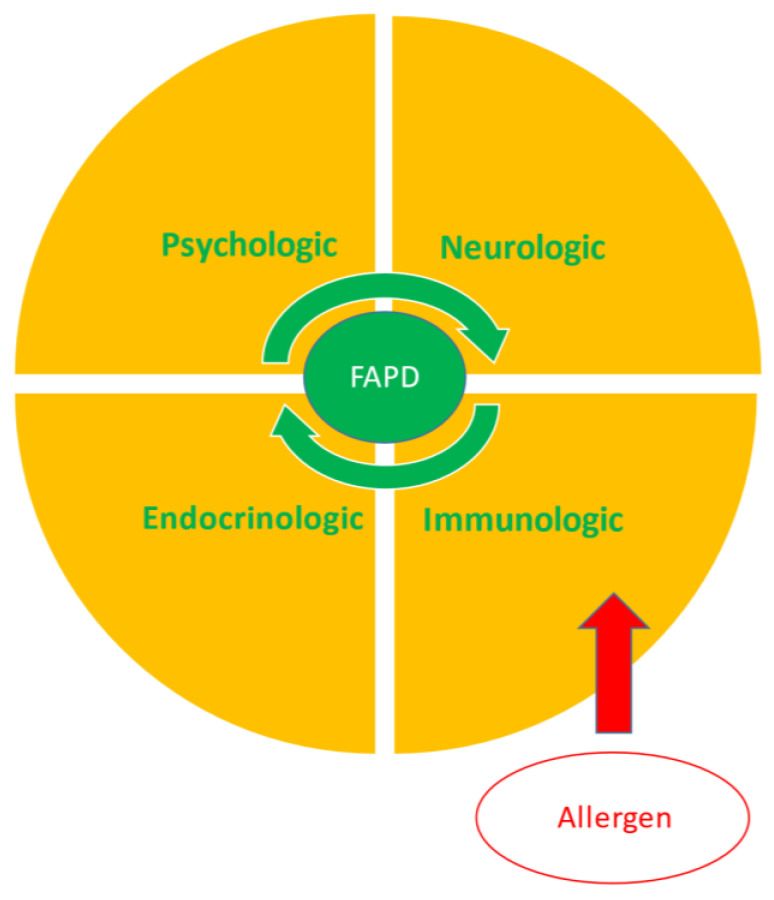
Four primary interacting systems generate the symptoms of functional abdominal pain disorders (FAPDs). This process can be initiated, maintained, or exacerbated by allergens through activation of the immune system.

**Figure 2 nutrients-13-02056-f002:**
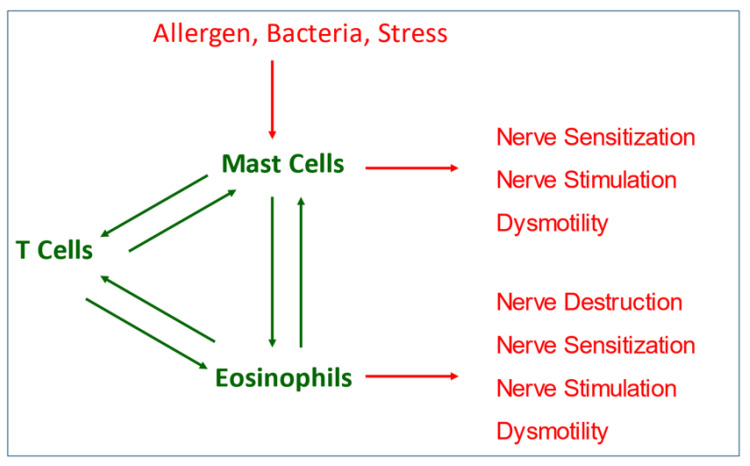
Within FAPD pathophysiology, particularly related to allergy, there are several therapeutic targets, including factors that stimulate or enhance an immunologic response, factors related to inflammatory cell infiltration or activation, or downstream effects following mediator release, either blocking receptors for released mediators or counteracting the physiologic effects of these mediators.

**Table 1 nutrients-13-02056-t001:** Trials in functional abdominal pain disorder patients utilizing medications with reported benefits in allergic conditions.

Medication	Mode of Action	Population	Study Type	Result
Diphenhydramine [129]	H1 antagonist	Adults with FD and mucosal mast cell density elevation	Open-label trial	Symptomatic improvement in 79%
Ebastine [128]	H1 antagonist	Adults with IBS	Randomized, double-blind placebo-controlled trial	Symptomatic improvement and reduced visceral sensitivity
Hydroxyzine/Ranitidine [130]	H1/H2 antagonists	Children with FD and mucosal eosinophilia	Retrospective case series	Symptomatic improvement in 50%
Loratidine/Ranitidine [131]	H1/H2 antagonists	Adults with FD	Retrospective case series	Symptomatic improvement in 71%
Montelukast [132]	Cys-Leukotriene antagonist	Children with FD and mucosal eosinophilia	Randomized, double-blind placebo-controlled cross-over trial	Superior to placebo in pain relief
Montelukast [133]	Cys-Leukotriene antagonist	Children with FD and mucosal eosinophilia	Open-label trial	Symptomatic improvement unrelated to changes in mucosal eosinophilia or mast cell density
Budesonide [137]	Steroid	Adults with FD and mucosal eosinophilia	Randomized, double-blind placebo-controlled trial	Symptomatic response not different from placebo
Unspecified PPI [135]	Proton pump inhibitor	Adults with FD and mucosal eosinophilia	Case-control study	Lower eosinophil density without symptomatic improvement
Pantoprazole [136]	Proton pump inhibitor	Adults with FD	Open-label trial	Symptomatic improvement and decreased mucosal eosinophil and mast cell densities

FD = functional dyspepsia; IBS = irritable bowel syndrome; PPI = proton pump inhibitor.

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
