# Peer review of "Update on the Role of Allergy in Pediatric Functional Abdominal Pain Disorders: A Clinical Perspective"

_nutrients, 2021, doi:10.3390/nu13062056_

Round 1
Reviewer 1 Report
This review discusses the role of allergy in pediatric FAPD. As this is an important and ‘hot’ topic which has gained a lot of attention in recent years, I would like to congratulate the authors for a nice overview which (in my opinion) could be improved further.
Major comments
Figure 1: endocrinological mechanisms are less clear and have not been discussed in detail.
Nutrient-induced hypersensitivity: important recent preclinical evidence that local IgE antibodies are involved in food-induced abdominal pain is missing (PMCID: PMC7610810).
The role of superantigens (as discussed in paper above) is also highly relevant as food allergy was also associated with S. aureus colonization in children with atopic dermatitis, please comment.
Related to (local) IgE, the authors of ref 75 suggest an atypical and non-IgE food allergy characterized by eosinophil activation, so please comment or correct the association with IgE on lines 190-1.
Although briefly mentioned in the introduction, the interaction between food and microbiota would warrant a more detailed description, especially for wheat (inactivation ATI, gluten) and FODMAP.
This interaction is also relevant in relation to the epithelium (doi: 10.3389/fimmu.2020.604054) and host factors, eg. proteases.
A(nother) figure of potential immune mechanisms is highly recommended for the reader. The same goes for treatment targets, as figure 1B of a previous publication (doi:10.4292/wjgpt.v4.i4.86)
A table showing FAPD/atopic overlap and pediatric (and adult) treatment studies would also help as an overview besides the text.
Minor comments
Lines 122-124: CRH-induced hyperpermeability with antigenic exposure is not demonstrated in this review (ref 36), so please cite original papers
Please check for double spaces and points instead of comma (eg. line 107).
Lunardi, 1991 is not referenced correctly (no number, line 304)
Reviewer 2 Report
Abdominal pain disorders are growing clinical entities, very complex. The physiopathologic mechanisms underlying are not well defined and interactions among environmental factors (diet, exposure to chemicals and pollutants), internal factors (specific immune response, psychological factors, etc) and microbiota are involved.
The review is useful and well organised. However I miss the complete absence in the review of an entity relate with FAPD. Food protein-induced enterocolitis syndrome (FPIES) and food protein-induced allergic proctocolitis (FPIAP). They are non IgE mediated food allergies that I believe should be consider in this review.
It is specially surprising the absence in the paragraph 3.2 "non IgE mediated Allergies" were discuss the possibility of a IgG mediated reaction and do not mention the possibility of T-cell mediated reactions and FPIES issue.
I encourage the authors to include a paragraph reviewing FPIES and FIAP and their relationship with abdominal pain in the same line done with food IgE allergies.
Round 2
Reviewer 1 Report
Thank you for clarifying the role of the endocrinologic system via HPA and CRH. However, it should be noted that many studies on CRH-immune interplay involve locally (and HPA-) produced CRH with eosinophil-mast cell interactions.
For the section on non-IgE mediated reactions: ref 89 is the same systematic review as ref 38 and please provide evidence from original paper for data on line 242-3 (“IL17 can induce eosinophil infiltration”).
Indeed, evidence cited (but not discussed) in the new ref 91 showed that small intestinal eosinophils actually suppressed Th17 cells (PMID: 26951334) so it is unclear how this fits with the allergic type and eosinophilic inflammation?
In this paper (ref 91), it is also unclear how duodenal Th17 cells relate to duodenal eosinophils (also in relation to controls, for which no duodenal data is shown if I’m not mistaken?) as arguably the duodenum has a more important role in FD as many studies have now shown.
Although I agree with moving the CLE-paper (ref 92) to non-IgE-mediated allergies, perhaps a role of local IgE (similar to the new ref 76) is also possible as this was not specifically measured: please comment.
Regarding interactions with bacteria, at least the potential role of proteases (either bacterial- or host-derived) deserves mentioning within food allergies (doi: 10.3389/fimmu.2020.604054)
Regarding the table, an overview of potential anti-allergic treatments (population, clinical effects, anti-inflammatory effects or mechanisms) would by very useful as the majority of papers are pediatric (and from the same group of the first author) with limited adult studies (eg PMID: 31040169, 34029411).
In this regards, authors should also consider adding evidence for anti-inflammatory effects of PPI as shown in cross-sectional (PMID: 29982192) and prospective studies (PMID: 33346007) in adults
Please check for other duplicate refs, eg refs 90 and 41.
